# Harnessing Pyroptosis for Cancer Immunotherapy

**DOI:** 10.3390/cells13040346

**Published:** 2024-02-16

**Authors:** Christopher M. Bourne, Cornelius Y. Taabazuing

**Affiliations:** Department of Biochemistry and Biophysics, Perelman School of Medicine, University of Pennsylvania, Philadelphia, PA 19104, USA; christopher.bourne@pennmedicine.upenn.edu

**Keywords:** pyroptosis, inflammasomes, cancer immunotherapy, antitumor immunity, tumor microenvironment, caspases, gasdermins, immune checkpoint blockade therapy

## Abstract

Cancer immunotherapy is a novel pillar of cancer treatment that harnesses the immune system to fight tumors and generally results in robust antitumor immunity. Although immunotherapy has achieved remarkable clinical success for some patients, many patients do not respond, underscoring the need to develop new strategies to promote antitumor immunity. Pyroptosis is an immunostimulatory type of regulated cell death that activates the innate immune system. A hallmark of pyroptosis is the release of intracellular contents such as cytokines, alarmins, and chemokines that can stimulate adaptive immune activation. Recent studies suggest that pyroptosis promotes antitumor immunity. Here, we review the mechanisms by which pyroptosis can be induced and highlight new strategies to induce pyroptosis in cancer cells for antitumor defense. We discuss how pyroptosis modulates the tumor microenvironment to stimulate adaptive immunity and promote antitumor immunity. We also suggest research areas to focus on for continued development of pyroptosis as an anticancer treatment. Pyroptosis-based anticancer therapies offer a promising new avenue for treating immunologically ‘cold’ tumors.

## 1. Introduction to Cancer Immunotherapy

One of the hallmarks of cancer is its ability to avoid immune detection and destruction [1]. Cancers utilize a variety of mechanisms to dampen, escape, and evade both adaptive and innate immunity [2,3,4]. Recently, new treatment strategies that rewire and stimulate the immune system to kill cancer cells have seen clinical success, such as targeted antibodies, checkpoint blockade, gene therapy, and chimeric antigen receptor (CAR) T cell therapy [5,6,7]. Each of these novel treatment strategies engages different arms of the immune system. Despite the clinical success of these therapies, there remains significant toxicities associated with certain treatments, such as CAR T cells, and only a fraction of patients respond to treatment with particularly poor response rates in solid tumors for some therapies, such as immune checkpoint blockade (ICB) therapy [8,9,10]. Thus, there is an urgent unmet need for new or complementary cancer immunotherapy strategies to be developed.

Pyroptosis has traditionally been thought of as an anti-pathogen form of cell death. However, because pyroptosis is an inflammatory type of cell death whereby intracellular contents are released into the extracellular milieu to induce a potent inflammatory microenvironment, there has been intense interest in harnessing this cell death modality for cancer immunotherapy applications. Mounting evidence, which has been extensively reviewed [11,12,13,14,15,16,17], suggests that combining pyroptosis with current immunotherapy agents leads to robust antitumor responses. Here, we discuss the latest innovative efforts to develop pyroptosis-based therapeutics as new or complementary anticancer treatments. We highlight how these therapies modulate the tumor microenvironment (TME) and stimulate the adaptive immune system to elicit antitumor responses. We conclude with suggestions about key focus areas for future studies that can aid in harnessing pyroptosis for cancer treatment.

## 2. Pyroptosis Overview

In the late 1990s and early 2000s, it had become apparent to researchers that enteropathogens such as *Shigella flexneri* and *Salmonella typhimurium* induced a programmed type of cell death that was distinct from apoptosis that resembled necrosis due to the pro-inflammatory nature of cell death [18,19,20]. However, rather than the seemingly accidental nature of necrosis, this cell death was programmed as it was dependent on caspase-1. Accordingly, Brad Cookson proposed the term pyroptosis, derived from the Greek word *pyro*, meaning fire or fever, and *ptosis*, which denotes falling, to describe the inflammatory nature of this cell death [20]. As such, pyroptosis was defined as caspase-1-dependent cell death until more recently.

In 2015, three landmark studies identified gasdermin D (GSDMD) as the long-awaited executioner of pyroptosis [21,22,23]. Normally, the N-terminus of GSDMD is inhibited by the C-terminus, but upon cleavage by caspase-1, the autoinhibitory C-terminal domain releases the N-terminus, which oligomerizes and forms pores in the plasma membrane [24,25,26]. GSDMD belongs to the gasdermin family of proteins, which were named based on the observation that they are primarily expressed in the gastrointestinal tract [27,28,29,30]. Mammals encode six of these (GSDMA, B, C, D, E, and PJVK (known as DFNB59)) [27,28,29,30]. Importantly, these studies revealed that in addition to caspase-1, caspases-4/11 can also cleave GSDMD to induce pyroptosis [21,22], prompting further research into the activation of other gasdermin proteins. It was soon discovered that caspase-3 cleaves GSDME (also known as DFNA5 (deafness, autosomal dominant 5)) and this results in GSDME N-terminal pore formation and pyroptosis, expanding the mechanism by which pyroptosis can be induced beyond caspase-1-mediated cell death [31,32]. Notably, all gasdermin proteins harbor N-terminal domains that can induce pyroptosis when cleaved by their respective proteases (described below) [30,33]. Appropriately, pyroptosis has now been redefined as gasderminmediated cell death [34,35], which encompasses the various mechanisms by which gasdermin proteins can be activated to induce pyroptosis.

## 3. Canonical Inflammasomes

Mammals utilize several germline-encoded pattern recognition receptors (PRRs) to detect conserved features of invading pathogens such as DNA and RNA that are collectively referred to as pathogen-associated molecular patterns (PAMPS) [36,37,38,39]. These PRRs can also detect host-derived danger signals known as damage-associated molecular patterns (DAMPS) [36,37,38,39]. Upon sensing PAMPS or DAMPS in the cytosol, the PRRs become activated and at least six are known to assemble into large multiprotein complexes called inflammasomes. Canonical inflammasome assembly usually involves recruitment of the ASC (apoptosis-associated speck-like protein containing a CARD) adaptor protein to the PRR, which then facilitates the recruitment of the pro-caspase-1 zymogen to form the inflammasome [36,37,39]. Pro-caspase-1 then undergoes autoproteolytic maturation to generate the active enzyme, which then cleaves and activates pro-inflammatory cytokines such as IL-1ꞵ and IL-18, as well as the pore forming protein GSDMD [40,41,42,43]. Cleavage of GSDMD induces oligomerization of the N-terminal domain that then forms pores in the plasma membrane to induce pyroptosis (Figure 1A) [21,22,23,24,25,26]. It is worth noting that in some instances, some PRRs can induce caspase-1 activation and subsequent GSDMD processing to induce pyroptosis independently of the ASC adaptor protein, but this typically occurs without efficient processing of cytokines [42,44]. Thus, activation of the canonical inflammasome pathway leads to caspase-1-dependent pyroptosis and depending on the composition of the inflammasome, may or may not include activation of IL-1ꞵ and IL-18, which can have important ramifications for the tissue microenvironment. However, even in the absence of active IL-1ꞵ and IL-18, other immunomodulatory molecules can potentially influence the microenvironment, such as HMGB1 (high-mobility group box 1 protein) and ATP (adenosine triphosphate), can still be released [12,14].

## 4. Non-Canonical Inflammasomes

Unlike the canonical pathway, which employs PRRs to detect danger signals, the non-canonical inflammasome pathway utilizes caspases-4/5 in humans and caspase-11 in mice to directly detect cytosolic LPS (lipopolysaccharide) from Gram-negative bacteria [21,22,36,45,46,47]. LPS binds directly to the non-canonical caspases (caspases-4/5/11) and facilitates their oligomerization, leading to their autoproteolytic maturation (Figure 1B) [48,49]. Interestingly, it was recently demonstrated that the formation of the caspase-11 non-canonical inflammasome requires caspase-11 catalytic activity, suggesting that initial dimerization and autoproteolytic activation facilitates oligomerization and speck formation as opposed to speck formation inducing caspase-11 autoproteolytic activation [50]. Active caspase-4/5/11 then cleave GSDMD to induce pyroptosis [21,22]. Formation of GSDMD pores results in potassium (K^+^) efflux and downstream activation of the canonical pathway via the NLRP3 inflammasome [51]. It was thought that downstream NLRP3 activation is required for cytokine activation by the non-canonical inflammasomes, but recent work demonstrates that the human non-canonical inflammasomes (caspases-4/5), but not the mouse (caspase-11), can directly cleave IL-18 to activate the cytokine [52,53,54], indicating that the non-canonical inflammasome pathway can contribute to inflammation independently of the canonical pathway (Figure 1B). Curiously, while caspase-11 cannot cleave IL-18, both the human and mouse non-canonical inflammasomes directly process IL-1ꞵ to deactivate the cytokine [54], suggesting that there are species specific similarities and differences with regard to non-canonical inflammasome signaling. However, the cleavage of GSDMD and induction of pyroptosis is conserved.

## 5. Inflammasome-Independent Pyroptosis

Inflammasomes specifically activate caspases-1/4/5/11, which then cleave and activate GSDMD to induce pyroptosis, but do not directly cleave other gasdermin family proteins. As such, other mechanisms exist to induce the processing of the other gasdermin proteins to induce pyroptosis. The host protease that activates human GSDMA currently remains unknown, but intriguingly, a recent study demonstrated that bird GSDMA is cleaved by bird caspase-1 in a tetrapeptide sequence dependent manner [55]. This suggests that human caspase-1 likely processed human GSDMA at one point but during evolution, human GSDMA evolved to lose the tetrapeptide sequence (YVHD) that confers specificity for caspase-1. A bacterial protease, SpeB, is the only known activator of human GSDMA [56,57], perhaps hinting that it has evolved to detect the activity of pathogenic proteases rather than host proteases. In contrast to GSDMA, the linker region of GSDMB, C, D, and E can be cleaved by caspase and granzyme proteases (Figure 1C–E).

There is significant crosstalk between the apoptotic and pyroptotic pathway [58], and the expression levels of the different gasdermin proteins dictate the mode of cell death. Consistent with this, the expression of gasdermin proteins converts apoptotic cell death to pyroptosis [59,60]. For example, it has been reported that GSDMC expression converts tumor necrosis factor alpha (TNFα)-induced apoptosis to pyroptosis by promoting caspase-8 activation, which subsequently cleaves GSDMC to induce pyroptosis (Figure 1C) [59]. Intriguingly, one study reports that αKG can lead to the formation of a caspase-8, GSDMC, and the death of the receptor DR6 complex, resulting in caspase-8-mediated processing of GSDMC and pyroptosis [61]. Generally, activation of caspase-8 results in the processing and activation of caspase-3, which can subsequently cleave and activate GSDME (if expressed) to induce pyroptosis (Figure 1C) [31]. Hence, activation of the apoptotic cascade can induce GSDMC or GSDME-mediated pyroptosis.

Cytotoxic lymphocytes (CTLs) such as T cells and Natural Killer (NK) cells inject granzyme proteases into target cells to mediate cell killing [60,62,63,64]. Granzymes are known to induce apoptosis but recent work demonstrated that granzyme A (GzmA) from T cells and NK cells directly cleaves GSDMB to induce pyroptosis (Figure 1D) [64]. In that study, none of the other granzymes tested could directly cleave any of the other gasdermin proteins (GSDMC, GSDMD, GSDME), but granzyme B (GzmB) was demonstrated to cleave and activate caspase-3, which subsequently processed and activated GSDME to induce pyroptosis (Figure 1D) [62]. Surprisingly, GzmB can also mediate caspase-independent pyroptosis by directly cleaving GSDME to induce pore formation [62]. In this manner, other proteases directly cleave gasdermin proteins to induce pyroptosis. In some instances, expression of the pore-forming N-terminal domains of the gasdermin proteins is sufficient to induce pyroptosis (Figure 1E). For instance, the N-terminus of mouse GSDMA3 and GSDMB have been reported to induce pyroptosis when delivered intracellularly (Figure 1E) [65,66]. Collectively, these findings suggest there may be alternative pathways in which apoptotic caspases and granzyme proteases mediate other cell death modalities; however, it is likely that the expression levels and kinetics by which the various effector proteins are cleaved and activated is what ultimately determines the cell death modality. Regardless, these findings highlight that there are various mechanisms to induce pyroptosis (discussed in more detail below) independently of the inflammasomes.

### Chemotherapeutics Induce CASP3/GDME-Mediated Pyroptosis

Chemotherapeutics, or the use of small-molecule drugs that block or engage biological pathways in cancer, has been the standard of care for many cancer types for decades. Traditionally, chemotherapeutics that disrupt oncogenic processes have been thought to induce apoptosis, but recently, new insights into the cell death pathways engaged by chemotherapies have expanded this view. In GSDME-expressing cells, chemotherapeutic agents induce caspase-3 (CASP3) activation, which then cleave GSDME to induce pyroptosis (Figure 1C) [31,60,67,68,69,70]. While normal cells retain GSDME expression, many cancer cells epigenetically downregulate the GSDME expression, or harbor a mutated GSDME protein that results in loss of function [60,71,72,73]. This suggests that cancer cells adapt mechanisms to actively inhibit pyroptotic cell death induced by chemotherapies to prevent immune effector cell engagement that would lead to antitumor immunity. Consistent with this idea, the expression of GSDME in GSDME low-expressing tumors converts chemotherapy-induced apoptosis into pyroptosis, indicating that the expression levels of the gasdermin proteins dictate the cell death modality [31,60,62,69,74]. Notably, this holds true for a variety of tumor models, including lung cancer, colon cancer, and breast cancer [60,67,68,69,70,74]. Importantly, recent studies demonstrated that an overexpression of GSDME in tumors without the addition of an exogenous activating signal was sufficient to induce pyroptosis and inhibit tumor cell growth [60]. Although it remains unclear what the activating signal was, the induction of pyroptosis elicited robust antitumor immunity that was dependent on CTLs, specifically, CD8^+^ T and NK cells [60]. Of note, GSDME-deficient tumors exhibited reduced levels of tumor-associated macrophage (TAMs) and tumor infiltrating lymphocytes (TILs) such as CD8^+^ T and NK cells, which expressed less perforin and GzmB, and had attenuated levels of IFNγ (interferon gamma) and TNFα compared to the GSDME-expressing tumors [60]. This suggests that the induction of pyroptosis promotes TIL activation and function to induce antitumor immunity. Consistent with this notion, cisplatin-induced GSDME-dependent pyroptosis increased the levels of TILs and cytokines associated with T cell activation (TNFα and IFNγ) in the tumor tissue [70]. Interestingly, despite the decrease in tumor growth, other immune cells, including B cells, dendritic cells, NK cells, and macrophages, were not recruited to the tumor tissue in this study [70].

Akin to other chemotherapeutics, a combination of BRAF and MEK inhibitors (BRAFi + MEKi), which activate caspase-3, have recently been reported to induce pyroptotic cell death in GSDME expressing cancers (Figure 1C) [67]. BRAFi + MEKi treatment led to GSDME-dependent tumor clearance and the establishment of antitumor immunity. This response was dependent on CD4^+^ and CD8^+^ T cells [67]. GSDME-expressing tumors exhibited increased intratumoral levels of CD4^+^ and CD8^+^ T cells and decreased TAMs and myeloid-derived suppressor cells (MDSCs) upon inhibitor treatment [67]. Notably, GSDME-deficient tumors had reduced T cell infiltration and tumor cell clearance but restoring GSDME processing resulted in pyroptosis and delayed tumor growth. Consistent with this, acquired resistance to BRAFi + MEKi was shown to be dependent on loss of GSDME cleavage, which correlates to a decrease in T cell infiltration in the tumor [67]. However, resistance can be overcome by using alternative chemotherapeutics such as etoposide to activate caspase-3 to cleave GSDME and induce pyroptosis [67]. These findings suggest that BRAFi + MEKi-resistant tumors lose T cell infiltration due to a loss of the ability to induce pyroptosis, which is immunogenic and required to generate antitumor immunity. In agreement with these findings, another study reported that genotype-matched treatment of cells with a MEK inhibitor (trametinib), an EGFR inhibitor (erlotinib), or ALK inhibitor (ceritinib), activated caspase-9, which subsequently induced caspase-3 activation and GSDME processing to mediate pyroptosis [69]. *In vivo,* both apoptosis and pyroptosis were reported to have contributed to effective tumor clearance [69], indicating that chemotherapeutics concurrently induce both pyroptosis and apoptosis. Interestingly, the antitumor effects of pyroptosis were diminished when the ability to undergo apoptosis was intact [69]. Several chemotherapeutics that induce pyroptosis have been reported [14] and future studies will pave the way for harnessing these compounds to induce pyroptosis for cancer immunotherapy.

## 6. CAR T Cell-Mediated Pyroptosis Induces Cytokine Release Syndrome

One of the most well-developed cancer immunotherapy strategies is the use of chimeric antigen receptor (CARs) T cells. CAR T cells are engineered ex vivo to localize to tumors and kill antigen-positive cancer cells. CARs are composed of an antigen-specific, single-chain variable fragment coupled to intracellular activation domains [75]. CAR T cell therapy has shown promise for treating B cell malignancies, but barriers limit their efficacy in solid tumors, such as antigen heterogeneity, immunosuppressive tumor microenvironment, and most importantly, toxicity [76]. CAR T cells kill target cells by injecting granzyme proteases to activate apoptosis. Given that granzymes can directly cleave gasdermin proteins to mediate pyroptosis, CAR T cells will induce pyroptosis in target cells if they express the appropriate gasdermins. Indeed, recent work indicates that GzmB from CAR T cells induces pyroptosis by activating caspase-3, leading to GSDME processing and pore formation (Figure 1D) [60]. However, rather than having a beneficial effect, this leads to cytokine release syndrome (CRS) in vitro and in vivo [62,77]. CRS results from activated macrophages that produce high levels of cytokines such as IL-1β and IL-6, which cause fever, respiratory insufficiency, and hypotension [76]. Mechanistically, Liu et al. reported that CAR T cells have higher levels of perforin and GzmB compared to existing CD8^+^ T cells, which allows CAR T cells, but not existing CD8^+^ T cells, to induce sufficient CASP3/GSDME processing to overcome membrane repair pathways and mediate pyroptosis [62]. ATP from the pyroptosing tumor cells subsequently activates the NLRP3 inflammasome in macrophages, resulting in caspase-1 activation and downstream GSDMD and IL-1β processing and release [62]. Additionally, HMGBI released during pyroptosis induces IL-6 production in macrophages, contributing to CRS [62]. Thus, the development of CAR T cell-mediated pyroptosis activators will prove challenging given the propensity to induce CRS. Perhaps this has hampered their development. Although CRS is reported for CAR T cell mediate therapy, this challenge is not unique to CAR T cells as other agents that induce pyroptosis also have the potential to release DAMPS that can cause CRS. Fortunately, one study suggests that CRS can be abated by using IL-1 blockade therapy such as anakinra [76]; however, the timing of therapy will have to be carefully considered to not dampen the beneficial immune activating effects of the cytokines released. Hence, more studies are needed to understand how CAR T cell-induced pyroptosis can be harnessed for immunotherapy applications.

## 7. DPP8/9 Inhibitors Induce Antitumor Immunity

Val-boroPro (VbP), also known as Talabostat or PT-100, was initially described for its ability to stimulate the immune system and induce potent antitumor responses; however, the mechanistic basis was unclear until recently [38,78,79]. VbP was initially thought to inhibit fibroblast activation protein (FAP), leading to reduced tumor growth in fibrosarcoma, lymphoma, melanoma, and mastocytoma-derived syngeneic-tumor models [78]. VbP treatment increased the expression of cytokines and chemokines known to promote the recruitment of innate immune effector cells as well as T cells. In line with this, the antitumor effects of VbP elicited immunological memory and involved tumor-specific CTLs. Immunodeficient mice exhibited a reduced response to treatment [78]. Later studies in syngeneic mouse tumor models confirmed that VbP required CD4^+^, CD8^+^, and CD11c^+^ cells to achieve maximal efficacy [79]. Importantly, adoptive transfer of T cells from VbP treated tumors was sufficient to reject tumor engraftment in naïve mice without the need for treatment [79]. Furthermore, as VbP-mediated tumor regression required dendritic cells, VbP synergized with dendritic cell vaccines and led to a complete regression of established tumors [79].

Based on the promising results in murine models, clinical trials were initiated for VbP [80,81,82,83]. With the knowledge that VbP engages innate immunity, VbP was tested alongside monoclonal antibody therapy, Rituximab, in a phase I clinical trial for patients with B cell malignancies [80]. Rituximab targets CD20 on B cells and B cell malignancies, directing professional antigen-presenting cells and NK cells to carry out ADCP (antibody-dependent cellular phagocytosis) and ADCC (antibody-dependent cell cytotoxicity) respectively. In this study, three out of twenty patients had a partial response and the majority of patients had elevated systemic cytokine levels [80]. VbP was also tested in phase II clinical trials, as both a single agent treatment for patients with metastatic colorectal cancer and in combination with cisplatin for patients with stage IV melanoma [81,82,83]. In the single agent trial, only two out of thirty-one patients responded to treatment, although one patient demonstrated a durable response [82,83]. In the combination trial with cisplatin, only six out of seventy-four patients responded to treatment [81,83]. A phase II clinical trial combining docetaxel with VbP in non-small-cell lung cancer (NSCLC) also observed promising antitumor benefits. A total of six out of fifty-five patients responded to treatment, two of which were complete responses [83]. On the basis of the phase I/II trials, two phase III trials were initiated for evaluating VbP in combination with docetaxel or pemetrexed for the treatment of NSCLC [83] but further development was stalled and VbP has not been FDA approved for the treatment of cancer.

It is important to mention that at the time of these clinical trials, the precise mechanism of action of VbP was unknown. Recent work has shown that VbP inhibits the serine hydrolases dipeptidyl peptidases 8 and 9 (DPP8/9), which activates the CARD8 and NLRP1 inflammasomes in humans and all functional alleles of the Nlrp1b inflammasome in mice (Figure 1A) [38,43,58,84,85,86,87]. Notably, CARD8 is the only canonical inflammasome that does not form an ASC-containing signaling platform, which is important for both caspase-1 and cytokine maturation (Figure 1A) [41,42,43,44]. Activation of the CARD8, NLRP1, and Nlrp1b inflammasomes leads to caspase-1 activation, which then cleaves GSDMD to induce pyroptosis [38,43,58,84,85,86,87]. Although VbP can inhibit other serine hydrolases such as FAP and DPP4, DPP8/9 are the specific targets that induce pyroptosis as DPP8/9-deficient cells are resistant to VbP-induced cell death [84,87]. Notably, Johnson and Taabazuing et al. demonstrated that VbP induced CARD8-dependent pyroptosis in human myeloid cells, including acute myeloid leukemia (AML) and B cell acute lymphoblastic leukemia (B-ALL) [84]. Consequently, DPP8/9 inhibitors were demonstrated to cause direct cytotoxicity of immortalized and primary AML cells in vivo in a caspase-1-dependent manner, resulting in a reduced tumor burden and an increased overall survival of the mice, suggesting that DPP8/9 inhibitors could represent a new treatment strategy for AML [84]. This study was one of the first to demonstrate that specifically inducing pyroptosis in cancer cells is a viable therapeutic treatment strategy for cancer, spurring further research into harnessing pyroptosis for immune-oncology purposes.

The recent mechanistic insights into the innate immune checkpoint regulated by DPP8/9 have reinvigorated interest in DPP inhibitors as anticancer agents [88,89,90]. Recent work demonstrated that a pan-DPP inhibitor (ARI-4175) increased the number of CD8 + T cells, which correlated with a decrease in hepatic lesions in a mouse model of hepatocellular carcinoma [90]. The antitumor effects were attributed to the activation of caspase-1 and induction of pyroptosis, but the inflammasome activation (whether NLRP1 or CARD8) remains unclear [90]. Another recent study reported VbP, synergized with gemcitabine, to induce pyroptosis and mediate tumor inhibition in a pancreatic cancer mouse model [88]. Similarly, VbP was reported to enhance T and NK cell infiltration into the tumor in a pancreatic ductal adenocarcinoma (PDAC) mouse model [89]. Moreover, VbP increased the efficacy of immune checkpoint blockade (anti-PD-1) therapy, which was dependent on both CD8^+^ and NK cells [89]. Notably, the combination of VbP and anti-PD-1 therapy elicited immunological memory, and some mice were protected from rechallenge [89], underscoring the promise of inducing pyroptosis as a standalone or complimentary therapy to current immunotherapy efforts.

Collectively, these findings suggests that DPP8/9 inhibitors may offer a unique opportunity to induce pyroptosis and unleash the host immune system to fight a variety of cancers. However, as these inhibitors can also cause pyroptosis in CTLs, their utility as immune-oncology agents will require further development [91,92]. Because the innate immune system appears to prime and augment the adaptive immune response to cancers when inflammasomes are activated, new approaches that combine both innate and adaptive immune therapies may offer the best chance at achieving clinical success. Some of these strategies are highlighted below.

## 8. Oncolytic Viruses Induce CASP3/GSDME-Mediate Pyroptosis to Establish Antitumor Immunity

Oncolytic viruses (OVs) are emerging as another method to induce pyroptosis for cancer immunotherapy applications [93,94]. As OVs can be modified to selectively target tumor cells and replicate in them, leading to tumor lysis and the release of tumor-specific antigens, the use of OVs represents a promising new immune-oncology strategy. In 2016, at least 40 clinical trials were recruiting patients to test the use of OVs as anticancer agents [93]. Based on the efficacious clinical outcomes and mild adverse events, an oncolytic herpesvirus therapy (Talimogene Laherparepvec (T-VEC)) has been approved by the FDA for the treatment of melanoma [95]. Notably, in pre-clinical models, T-VEC was synergistic with anti-CTLA-4 (cytotoxic T-lymphocyte antigen 4) ICB therapy, resulting in an increase in tumor-specific CD3^+^ and CD8^+^ T cells [95,96]. While an increase in neutrophils, monocytes, and chemokines was also observed, T cells appeared to play a major role as the depletion of CD8^+^ T cells abolished both systemic and local efficacy. Due to the efficacy and safety profile of OV therapy, there is much interest in further developing oncolytic virotherapy.

### 8.1. A Nanoprodrug Combined with Herpes Simplex Virus 1 Induces Antitumor Immunity

Several reports indicate that OVs activate the CASP3/GSDME axis to induce pyroptosis (Figure 1C) [97,98,99,100,101,102]. Su et al. developed a nanoprodrug + OV approach for cancer immunotherapy [100]. They loaded Niclosamide, an FDA approved STAT3 (signal transducer and activation of transcription 3) inhibitor into self-assembling, pre-engineered prodrug nanoparticles, which they dubbed MPNPs, that are ROS/pH responsive to trigger the drug release after endocytosis. The MPNPs can induce tumor cell pyroptosis on their own via the CASP3/GSDME axis, but maximum efficacy was achieved when combined with oncolytic herpes simplex virus 1 (oHSV). MPNPs + oHSV treatment resulted in CASP3/GSDME-mediated tumor cell pyroptosis, which inhibited tumor growth in a 4T1 TNBC model. Notably, the TME was remodeled to promote antitumor immunity. Specifically, there was an increase in the effector memory T cells (T_EM_), central memory T cells (T_CM_), and mature dendritic cells, as well as a decrease in regulatory T cells (Tregs) and MDSCs. Consistent with the antitumor promoting immune landscape, MPNPs + oHSV treatment prevented metastasis and tumor recurrence. The depletion of CD4^+^ and CD8^+^ T cells suggests that both are important but CD8^+^ T cells played a more significant role in establishing antitumor immunity in this context. Of note, MPNPs + oHSV treatment was synergistic with anti-PD-1 therapy, indicating that the MNPs + oHSV treatment converted an immunologically ‘cold’ TME into a ‘hot’ TME. Thus, the use of OVs represents a promising new strategy to sensitize cells to ICB therapy.

### 8.2. Vesicular Stomatis Virus Induces Immunogenic Cell Death

Like HSV-1, vesicular stomatitis virus (VSV) therapy can overcome ICB resistance [97]. Lin et al. demonstrated that VSV treatment induced CASP3/GSDME-dependent pyroptosis, which inhibited tumor growth in a B16F10 melanoma model [97]. Consistent with the reports using other OV-based therapies, they observed an increase in CD3^+^CD8^+^ T cells upon VSV treatment. Importantly, GSMDE-deficient B16F10 cells failed to robustly recruit CTLs, indicating that tumor cell pyroptosis is critical for establishing antitumor immunity. It is worth mentioning that GSDME-deficient tumors still retain the ability to undergo apoptosis. This suggests that tumor cell pyroptosis is critical for establishing antitumor immunity, likely due to the release of tumor-specific antigens that promote adaptive immune activation. Like other OV-based therapies, VSV treatment enhanced the efficacy of ICB therapy, indicating that OVs induce immunogenic cell death.

### 8.3. A Recombinant Adeno-associated Virus Delivers GSDMD N-terminus to Induce Tumor Cell Pyroptosis

Recently, Lu et al. developed a recombinant adeno-associated virus (rAAV) expressing the N-terminus of GSDMD to treat a variety of tumors [99]. To avoid the toxicity of GSDMD N-terminus when generating virus, the authors packaged the virus in insect cells and used a human promoter that drives the expression of GSDMD N-term in mammalian cells that exhibited low activity in insect cells. In an *in vivo* glioblastoma model, treatment with rAAV inhibited tumor growth compared to the control-treated mice. Strikingly, they observed a transient opening of the blood brain barrier, which the authors postulate facilitated the infiltration of CTLs to eliminate the tumor. This rAAV therapy was also efficacious in a 4T1 TNBC model. They observed tumor cell pyroptosis that resulted in tumor growth inhibition. This effect was likely mediated by the immune system as they observed an increase in CD3^+^, CD4^+^, CD8^+^ T cells, and NK cells at the tumor sites. Indeed, no tumor regression was observed upon rAAV treatment in athymic Nu/Nu mice, which lack mature T cells. Of note, the rAAV therapy was synergistic with anti-PD-L1 therapy in the TNBC model, suggesting that OVs may work as monotherapies or in combination with other cancer immunotherapies for solid tumors.

### 8.4. Coxsackievirus Group B3 Induces CASP3/GSDME-mediated Pyroptosis

Coxsackievirus is an RNA virus that is thought to have oncolytic activity against a wide variety of cancers, including breast cancer, lung cancer, endometrial cancer, and colon cancer [102,103,104,105]. Zhang et al. recently reported that wildtype coxsackievirus group B3 (CVB3) induced CASP3/GSDME-mediated pyroptosis to kill cancer cells [102]. CVB3 infection increased the levels of reactive oxygen species (ROS), which activated CASP9 that subsequently activated CASP3. Of note, they and others reported that no CASP1 and GSDMD activation was observed with OV infection [97,102]. In vivo, CVB3 reduced colon cancer cell growth in a nude mouse model. However, there was a slight decrease in body weight, suggesting there may be some toxicity associated with CVB3 treatment. The authors observed CASP3 and GSDME cleavage of tumor cells in vivo but they did not profile the immune landscape to determine the impact on adaptive immunity and the establishment of antitumor immunity. Thus, future studies are needed to determine the therapeutic potential of CVB3 in establishing antitumor immunity.

### 8.5. Parapoxvirus Ovis Induces CASP3/GSDME-mediated Tumor Cell Pyroptosis to Elicit Antitumor Immunity

Like other OVs, oncolytic parapoxvirus ovis (ORFV) was initially thought to induce apoptosis, but recent work demonstrates that ORFV induces GSDME-mediated pyroptosis to elicit antitumor immunity [98]. Lin et al. reported that the treatment of EMT6 breast cancer cells with ORFV-induced CASP3/GSDME-mediated pyroptosis, which converts to apoptosis in GSDME-deficient cells [98]. Intriguingly, they observed that ORFV treatment increased the mRNA expression of pro-inflammatory genes, including *Gsdmd, Il1b, Casp1, Nlrc4, and Casp8.* Strikingly, ORFV treatment also stabilized the GSDME protein by decreasing GSDME ubiquitination, preventing its proteasomal degradation. Accordingly, while etoposide was unable to induce pyroptosis in GSDME-low expressing tumors (B16F10 and 4T1), ORFV treatment induced pyroptosis, resulting in inhibition of tumor growth in vivo. Importantly, ORFV also induced pyroptosis in primary patient tumors treated ex vivo. As expected, ORFV treatment increased the numbers of CD3^+^ and CD8^+^ T cells, antigen-specific CD8^+^ T cells, GzmB^+^CD8^+^ T cells, and NK cells. Critically, the recruitment of TILs was significantly attenuated in GSDME-deficient tumors. Furthermore, GSDME-deficient B16F10 tumors treated with ORFV had increased metastasis compared to wildtype cells. It is worth mentioning that GSDME-deficient tumors were demonstrated to undergo apoptosis. Depletion of CD8^+^ T cells prevented ORFV-mediated tumor growth inhibition. These data strongly imply that tumor cell pyroptosis is required to drive the recruitment of TILs that generate antitumor immunity. Importantly, recombinant ORFV that expressed enhanced green fluorescent protein could induce pyroptosis to the same degree as unmodified ORFV, suggesting that OVs are efficacious even when modified to the expressed desired proteins. Consistently, they observed tumor-specific replication of recombinant ORFV that was administered systemically, which inhibited B16F10 metastasis to the lungs.

Because chemotherapeutics are known to induce pyroptosis in GSDME expressing tumors and ORFV treatment stabilizes GSDME protein, the authors explored a combination therapy with etoposide + ORFV. They observed increased pyroptotic death with the combination therapy compared to ORFV monotherapy. This coincided with an increased GzmB^+^CD8^+^ T cell infiltration in the combination therapy, prompting them to test a triple-treatment with anti-PD-1 therapy. As expected, the triple treatment (etoposide + ORFV + anti-PD-1) displayed an additive effect in a 4T1 TNBC model and led to an extension in overall survival of treated mice. These data reinforce the notion that tumor cell pyroptosis can promote the recruitment of TILs that modulate the TME to sensitize immunologically ‘cold’ tumors to ICB therapy and induce antitumor immunity.

### 8.6. Recombinant Measles Virus Induces BAX/BAK- and CASP3/GSDME-Dependent Pyroptosis

Wu et al. recently reported that recombinant measles virus based off the Chinese measles vaccine strain Hu191 (rMV-Hu191) induced CASP3/GSDME-mediated pyroptosis in esophageal squamous cell carcinoma (ESCC) to facilitate antitumor immunity [101]. They demonstrated that rMV-Hu191-mediated proptosis was dependent on BAX/BAK, CASP3, and GSDME in ESCC cell lines. They then tested the efficacy of rMV-Hu191 therapy *in vivo* and observed that in immunodeficient mice, rMV-Hu191 treatment delayed tumor growth and extended the survival of the mice compared to the control group. Importantly, there was no toxicity associated with rMV-Hu191 treatment, as assessed by body weight measurements, serum alanine transaminase, aspartate aminotransferase, and creatine levels. Although immune competent mice were not tested, based on the observation that tumor cell pyroptosis occurred with rMV-Hu191 treatment, this therapy would be expected to induce antitumor immunity.

Taken together, the use of OVs offers a novel strategy to harness the immunogenic nature of pyroptotic cell death to mediate antitumor immunity. The precise mechanism of how OVs induce pyroptosis remains to be fully elucidated. However, the data suggest that OVs activate the intrinsic apoptotic pathway by disrupting mitochondrial stability and causing BAX/BAK-mediated release of cytochrome c, leading to apoptosome formation and CASP9 activation. Active CASP9 then activates CASP3, which subsequently cleaves GSDME to induce pyroptosis. Importantly, pyroptosis promotes the recruitment of innate and adaptive immune cells to the tumor site to mediate antitumor immunity. Thus, although CTLs can induce pyroptosis, it appears that tumor cell-mediated pyroptosis (as opposed to immune cell-mediated pyroptosis) is required for the induction of antitumor immunity since tumor cell pyroptosis facilitates the recruitment of immune cells to the tumor site. However, a contribution from immune cell-mediated pyroptosis in establishing antitumor immunity cannot be ruled out. It is also possible that infiltrating immune cells at the tumor sites undergo pyroptosis and release cytokines that mediate antitumor immunity. Future studies are needed to delineate the contributions from tumor cell-mediated pyroptosis and immune cell-mediated pyroptosis in establishing antitumor immunity. Notably, the degree to which the tumor cells undergo pyroptosis appears to influence the robustness of antitumor immunity given that further synergy was achieved with a triple therapy (etoposide + ORFV + anti-PD-1) compared to a mono or dual therapy. Altogether, oncolytic virotherapy is an encouraging new treatment modality that warrants further development. A major hurdle that may need to be overcome is the fact that patients may generate neutralizing antibodies against OVs, limiting their repeated use.

## 9. Next-Generation Approaches to Inducing Pyroptosis for Anticancer Applications

### 9.1. An Epigenetics-Based Approach for Enhancing Chemotherapy-Induced Pyroptosis

Early on, many of the therapeutics that induced pyroptosis in cancer were determined after their initial development. As understanding of the mechanisms of pyroptosis and their potential therapeutic benefits has progressed, researchers have begun designing therapeutics specifically to induce pyroptotic cell death in tumors. Most of these seem to be nanoparticle-based approaches that allow for somewhat specific tumor targeting. For example, Fan et al. used decitabine, a DNA methyltransferase inhibitor that promotes DNA hypomethylation to epigenetically upregulate GSDME expression, followed by delivering a tumor-specific nano-liposome packaged with cisplatin to induce pyroptosis in a 4T1 TNBC model [106]. This treatment regimen led to an increased pyroptosis compared to the cisplatin treatment alone. Importantly, it resulted in an increased infiltration of the CTLs (CD3^+,^ CD8^+^) in the tumor environment, as well as increased dendritic cell maturation (CD11c^+^,CD80^+^,CD86^+^) in the draining lymph nodes [106]. Furthermore, they observed a shift in CD8^+^ T cells into central memory CD8^+^ T cells in the spleen along with decreased metastasis, suggesting that protective immunity was achieved [106].

### 9.2. A Nano-CRISPR Scaffold for Inducing Pyroptosis

In a similar strategy, Wang et al. also applied insights from the CASP3/GSDME axis to create a nanoplatform for activating both effectors (Figure 1) [107]. The nanoplatform (called NanoCD) comprised a nanoparticle co-loaded with cisplatin and catalytically inactive Cas9 fused to a transcription activator (CRISPR/dCas9) that was designed to release the cargo upon acidification in the endosome. When released, the dCas9, which is fused to a transcription activator, contains a guide RNA that localizes it to the endogenous GSDME locus where it induces the expression of the GSDME. Subsequently, GSDME is cleaved by active CASP3, which has been activated by the cisplatin released in the cytosol, to induce pyroptosis. In vitro, NanoCD-treated B16F10 cells released DAMPs associated with anti-tumor immunity, including uric acid, ATP, calreticulin (CRT), and HMGB1 [107]. Using co-culture assays, they demonstrated that NanoCD treated cells induced activation of dendritic cells, as assessed by increased TNF-α, IFNγ, IL12p40 secretion and an increased expression of the T cell costimulatory ligands CD40, CD80, and CD86 [107]. In vivo, NanoCD showed efficacy against the B16F10 melanoma model through increased recruitment of CD4^+^ and CD8^+^ T cells, and reduced recruitment of the immunosuppressive CD206^+^ macrophage population and Tregs [107]. While elevated levels of IL-12 and IFNγ promote immunity, they also reported increased TGF-β and IL-10, which are thought to be immunosuppressive cytokines [108]. NanoCD strongly synergized with anti-PD-1 therapy, demonstrating more efficacy through further activation and recruitment of immune cells [107]. Importantly, the combined treatment of NanoCD and anti-PD-1 markedly increased the effector memory T cell population and protected the mice from recurrence and metastasis. Interestingly, both in vitro and in vivo, the authors observed an increase in IL-1β levels, but a mechanistic link between caspase-3 activation and IL-1β maturation was not explored. It remains possible that the K^+^ efflux mediated by GSDME pores can activate the NLRP3 inflammasome downstream of caspase-3 activation, which would activate caspase-1 to induce IL-1β maturation [51,109]. However, it could be that GSDME-mediated pyroptosis occurs too rapidly to permit cells to complete their ‘bucket list’ [110], such as downstream caspase-1 activation, before their eventual demise. Thus, whether the IL-1β released during GSDME-mediated pyroptosis is proteolytically matured or not remains unclear and warrants further investigation as this would reveal important insights into the tumor microenvironment. Notably, active IL-1β was reported to be important for eliciting antitumor immunity during radiation therapy as inflammasome deficient mice failed to mediate tumor destruction [111].

### 9.3. A Biorthogonal System for Delivering GSDMA3 Protein

Another strategy that has proven effective in eliciting antitumor immunity involved the generation of a biorthogonal system capable of releasing GSDMA3 protein to induce pyroptosis in cancer cells (Figure 1E) [66]. The authors conjugated GSDMA3 to a gold nanoparticle (NP-GSDMA3) using a silyl ether linkage that can undergo desilylation to release the free protein in the presence of phenylalanine trifluoroborate (Phe-B_3_) [66]. Both intravenous and intratumoral injections of NP-GSDMA3 and Phe-BF_3_ showed good uptake of the nanoparticles at the tumor site in a 4T1 tumor model, coincident with similar levels of tumor regression that was dependent on an intact immune system as mice lacking mature T cells did not show tumor regression with treatment. Accordingly, treatment with NP-GSDMA3 and Phe-BF_3_ resulted in an increase in the CD3^+^, CD4^+^, and CD8^+^ T cells population at the tumor site, along with a decrease in the Foxp3^+^ Treg population. Furthermore, antibodymediated depletion of either CD4^+^ or CD8^+^ T cells abrogated NP-GSDMA3 treatment-induced tumor regression, indicating that tumor regression due to pyroptosis required both CD4^+^ T helper cells as well as cytotoxic CD8^+^ T cells. Using single-cell RNA sequencing, the authors found that NP-GSDMA3 and Phe-BF_3_-treated tumors had increased CD4^+^, CD8^+^, and NK cell populations, but decreased monocytes, myeloid derived suppressor cells, and neutrophil cell populations. Furthermore, the M1 polarized macrophage population increased while the M2 polarized population decreased, consistent with a cytotoxic TME. Notably, they observed an upregulation of genes that promote chemotactic cytokines, lymphocyte activation, and antitumor effector genes such as the granzymes (*Gzma, GzmB, Gzmk)* and a decrease in immunosuppressive genes, including those that encode for programmed death ligand 1 (PDL-1) and PDL-2 in treated mice, suggesting that this treatment modality would synergize with anti-PD-1 ICB therapy. Importantly, the 4T1 tumor is considered immunologically ‘cold’, and as such, did not respond to anti-PD-1 therapy alone. However, a single treatment with the NP-GSDMA3 and Phe-BF_3_, which does not inhibit tumor growth on its own, followed by anti-PD-1 therapy, caused a significant shrinkage in tumor volume, suggesting the immunologically ‘cold’ immune checkpoint resistant TME was converted to a ‘hot’ TME via the induction of pyroptosis. Their data suggest that only 15% of the cells need to pyroptose to induce a robust antitumor response that leads to tumor regression. Strikingly, the levels of HMGB1, IL-18, and IL-1β were all elevated in the serum of treated mice and while an anti-IL-1β antibody markedly inhibited tumor regression, anti-IL-18 and anti-HMGB1 antibodies exhibited only a partial- and non-inhibition of tumor regression, respectively, indicating that IL-1β plays a significant role in promoting antitumor immunity. However, whether IL-1β and IL-18 were processed into the mature form remains unclear.

### 9.4. An Attenuated Bacterium Shuttles GSDMD into Cells to Induce Pyroptosis

During pyroptosis, the GSDMD membrane pores can be repaired by the endosomal sorting complex required for transport (ESCRT) in a calcium (Ca^2+^) influx-dependent manner to inhibit cell lysis [112]. Thus, Li et al. reasoned that they could enhance GSDMD-mediated pyroptosis and antitumor response by inhibiting Ca^2+^ influx to prevent membrane repair [113]. To test this, they synthesized glutathione (GSH)-responsive GSDMD protein cages that were designed to be released preferentially in the GSH-rich tumor cells and tethered the cages to an attenuated *Salmonella typhimurium* (VNP) strain (VNP-GD). Once intracellular, the bacterial-tethered GSDMD was released and was cleaved by active caspase-1, which is activated by the bacteria, to induce pyroptosis. To prevent Ca^2+^ influx and membrane repair, they co-treated with a nanoparticle encapsulated with a calcium chelator (BAPTA-AM). In a 4T1 breast and B16F10 melanoma mouse model, they observed that inclusion of the Ca^2+^ influx antagonist was more efficacious at preventing tumor growth compared to the VNP-GD-only treated mice. As expected, tumor regression was further enhanced when combined with anti-PD-1 therapy. They profiled the immune landscape and observed increased dendritic cell maturation (CD80^+^,CD86^+^), increased CD8^+^ T cell infiltration and activation, as assessed by levels of TNFα and INFγ, as well as an increased HMGBI expression in treated tumors. Treated mice also exhibited a decreased metastasis and reduction in distant tumor implants following treatment of only the primary tumors, consistent with the induction of antitumor immunity following pyroptosis. Lastly, the authors also showed that this treatment modality was efficacious in an inoperable ovarian cancer model. Collectively, these findings suggest that inhibiting the ESCRT-mediated membrane-repair pathway can augment pyroptosis-based cancer treatments.

### 9.5. Self-Assembling Nanovaccines Activate the NLRP3 Inflammasome

A major challenge in cancer immunotherapy development is the low immunogenicity and lack of neoantigens from certain cancers. To overcome this, Manna et al. developed a nanovaccine approach that utilizes a potent toll-like receptor 7/8 (TLR7/8) agonist in combination with a peptide that activates the NLRP3 inflammasome that self-assembles into nanoparticles that can be readily formulated with antigens targeting specific cancers [114]. Using this approach, they demonstrated that their nanovaccine led to NLRP3-dependent pyroptosis, which resulted in increased CD4^+^ and CD8^+^ T cell activation in an E.G7-OVA lymphoma model. Furthermore, they showed that their nanovaccine synergized with a checkpoint blockade (anti-CTLA-4 + anti-PD-L1 cocktail) to block tumor growth in B16F10 melanoma and CT26 colon cancer mouse models. The combined treatment enhanced CD4^+^ and CD8^+^ TILs without increasing Foxp3^+^ Tregs. Of note, when surviving mice in the treated group were rechallenged with CT26 cells, all mice rejected the new tumor growth, indicating that the mice developed immunological memory. This strategy holds promise as it promotes antigen-specific T cell responses. Moreover, as this involves activation of the canonical NLRP3 inflammasome, molecules important for innate immune defenses that need to be proteolytically matured, such as IL-1β and IL-18, would be biologically active and promote adaptive immune-mediated anticancer immunity [115,116,117,118,119].

### 9.6. Photodynamic Therapy Induces Pyroptosis

Recently, photodynamic therapy (PDT) has also been demonstrated to induce pyroptosis and elicit antitumor immunity [120]. A carrier-free chemo-photodynamic nanoplatform (A-C/NP) that triggered the CASP3/GSDME-mediated pyroptosis was developed [120]. Supernatants from 4T1 breast cancer cells treated with A-C/NPs and laser light to induce pyroptosis promoted CD4^+^ and CD8^+^ T cell activation in vitro. Consistent with the in vitro results, A-C/NP + light also induced pyroptosis in vivo, which induced abscopal effects in a bilateral subcutaneous 4T1 mouse model and inhibited tumor recurrence. As anticipated, they observed an increase in CD4^+^ and CD8^+^ T cells in the spleen and at distant tumor tissues, indicative of a systemic immune response. Interestingly, they noted elevated levels of NLRP3, IL-18, and IL-1β in the serum of treated mice, but whether the NLRP3 inflammasome or the cytokines contributed to the immune response was not investigated.

### 9.7. GSDMB N-Term Encoding mRNA Lipid Nanoparticles Induce Pyroptosis

The recent success of the COVID-19 lipid nanoparticle vaccine demonstrated that gene therapy can be efficacious and scalable through lipid nanoparticles. Researchers are already working to apply these favorable findings to generate promising cancer immunotherapies and immune adjuvants. Towards this, lipid nanoparticles with mRNA encoding the N-terminus of Gasdermin B were generated [65]. Empirically, the N-terminus of Gasdermin B, as compared to the other Gasdermin proteins, was found to induce the highest levels of cell death in a panel of cancer cell lines, such as human 293T and HeLa cells, and murine 4T1 and B16F10 cells. GSDMB-N LNPs induced the release of lactate dehydrogenase, HMGB1, and ATP from cancer cells, which stimulated TNFα, IFNγ, IL12, and IFNβ secretion from bone marrow-derived dendritic cells. These nanoparticles were efficacious against 4T1 and B16F10 melanoma, with B16F10 responding stronger. Antitumor effects were attributed to released intracellular DAMPs as well as IL-1β and IL18. Pyroptosis induced by GSDMB-N is caspase-1-independent and would thus not be expected to yield mature IL-1β or IL-18 unless NLRP3 was activated downstream. Hence, the precise mechanism of pyroptosis and the signaling capabilities of cytokines present in the TME warrant further investigation. Systemically, mice treated with these LNPs had increased levels of TNFα and IFNγ, and CD8^+^ T cell recruitment in the tumors. Additionally, in the B16F10 model, LNP-treated mice had increased CD11c^+^ MHCII^+^ activated dendritic cells (DCs), CD4^+^ and CD8^+^ T cells, NK cells, and inflammatory monocytes. Strikingly, GSDMB-N LNPs demonstrated abscopal immune effects, whereby one treated flank tumor induced immune responses against the untreated tumor on the other flank. As anticipated, these antitumor effects synergized with an anti-PD-1 checkpoint blockade. These studies employed intratumoral delivery of the nanoparticles. While intratumoral injections of LNPs may not be clinically feasible in all cases, a targeted approach could allow for specific localization of pyroptotic instructions within the tumor [65].

The above-described studies represent some of the innovative strategies that are being explored to harness the immunogenic nature of pyroptotic cell death for cancer immunotherapy applications. This is a burgeoning area of research and several studies have been undertaken that cannot all be covered in this review. However, these findings collectively indicate that the induction of pyroptosis holds tremendous potential as a single agent or complimentary anti-cancer therapy. Even if ineffective as a single agent treatment, it rewires the TME such that ‘cold’ tumors become ‘hot’ and are responsive to ICB therapy. Critically, the molecules released by pyroptotic cells and their impact in modulating the TME need to be clearly defined to advance this therapeutic approach. Some of these molecules and their impact are described below. A summary of the pyroptosis-based strategies, tumor models, and impact on adaptive immunity is summarized in Table 1.

## 10. Cytokines Released during Pyroptosis Have Context-dependent Roles in Cancer Initiation and Progression

The induction of pyroptosis can be a double-edged sword in the context of cancer. For example, GSDME-mediated pyroptosis has antitumor effects when triggered by chemotherapeutics, but when triggered by CAR T cells, it results in CRS. Chronic inflammation resulting from aberrant inflammasome activation, infection, diet, or other factors can promote tumor initiation, progression, and metastasis [134]. Conversely, as demonstrated by the ability of pyroptosis-based therapies to eliminate tumors, acute inflammation can have beneficial antitumor effects. Hence, understanding the impact of inflammation on cancer initiation, metastasis, and response to treatment is an active area of research. As efforts to induce pyroptosis for cancer immunotherapy represent a promising anticancer strategy, understanding the biological impact of IL-1β and IL-18, and the cytokines processed and released during pyroptosis will be key to understanding how inflammation influences cancer progression and treatment. The effects appear to be pleotropic with many studies suggesting that IL-18 and IL-1β exhibit both pro-tumorigenic and anti-tumorigenic properties, highlighting the complex role of inflammation in tumorigenesis. IL-1ꞵ is largely thought to promote tumor growth, and the vast majority of IL-1β-based therapies aim to block the cytokine and its signaling, while IL-18 is largely thought to promote antitumor immune responses, and many therapeutic efforts aim to boost IL-18 signaling.

### 10.1. IL-1β Both Promotes and Inhibits Tumor Initiation and Progression

IL-1ꞵ regulates diverse immunological processes such as immune cell recruitment and activation, angiogenesis, and cell fate changes [116,134,135]. Additionally, IL-1β activates the NF-kB signaling pathway, which can promote cell survival by expressing antiapoptotic factors [134,136]. In cancer, IL-1ꞵ has been shown to have both antitumor and pro-tumorigenic functions, with more evidence suggesting a pro-tumorigenic role. Chronic IL-1ꞵ signaling promotes tumorigenesis by activating endothelial cells, promoting tumor angiogenesis, and recruiting immunosuppressive cells [134]. In agreement, late stage melanoma patients exhibited high levels of IL-1β in their serum and presented with increased MDSCs and Tregs [121]. Notably, this correlated with a poor prognosis and decreased progression-free survival, suggesting that chronic inflammation induced suppressive immune cells that support tumor growth in advanced melanoma patients [121]. Also in support of the notion that IL-1β promotes tumorigenesis, gain-of-function mutations in NLRP1 caused constitutive activation and secretion of IL-1β in keratinocytes, resulting in skin inflammation that made the patients more susceptible to skin cancer [137]. Similarly, in B16F10 melanoma, the inhibition of NLRP3 reduced tumor growth and synergized with checkpoint blockade therapy, suggesting that inflammasome-derived IL-1β promotes tumorigenesis [138]. IL-1β derived from tumor-associated macrophages was also reported to promote tumorigenesis and blocking IL-1β led to tumor control [122]. While much evidence points to a negative role of IL-1β on tumor initiation and progression, these studies largely rely on chronic inflammation models.

Due to the prevailing thought that IL-1β promotes tumor progression, many clinical trials aim to block IL-1β signaling as an anticancer therapy. Furthermore, many of these clinical trials examine the IL-1β blockade with other immunotherapies such as checkpoint blockade [139]. Of note, in a large clinical trial, an IL-1β blocking antibody (canakinumab) significantly reduced the incidence of lung cancer and improved overall survival, suggesting a pro-tumorigenic role for IL-1β in lung cancer [123]. A follow-up trial in lung cancer using canakinumab in combination with radiation or chemotherapy failed to demonstrate efficacy, implying that IL-1β blockade may be beneficial for preventing cancer initiation but does not augment treatment [124]. In fact, given that treatment-induced IL-1β seems to promote tumor clearance, IL-1β antagonists during treatment may be detrimental and attenuate antitumor adaptive immune responses. An IL-1β-blocking antibody (anakinra) was also reported to prevent metastasis in a humanized mouse model of breast cancer [140,141]. In addition to direct IL-1β antagonists, anticancer treatments that target upstream events in the inflammasome pathway, specifically, NLRP3 inhibitors, are also being developed as antitumor agents [142].

Although many of the therapies are focused on inhibiting IL-1β, as described above, treatment-induced IL-1β production elicits antitumor effects. IL-1β, as opposed to IL-18, may play a dominant role in mediating anti-tumor immunity [66]. In support of this, it was recently reported that inflammasome activation in dendritic cells induces a hyperactive state in a subset of dendritic cells that have an enhanced ability to migrate to draining lymph nodes and stimulate potent CTL responses [125]. Importantly, the hyperactive phenotype is dependent on the inflammasome-derived IL-1β, which promotes the eradication of tumors that are resistant to anti-PD-1 therapy [125]. Thus, IL-1β has context-specific roles in tumorigenesis; however, the evidence suggests that IL-1β generated by pyroptosing cancer cells promotes antitumor immunity.

### 10.2. IL-1α Has Pleotropic Effects on Tumorigenesis

IL-1α is constitutively expressed by many cell types and shares the same receptor as IL-1β. However, unlike IL-1β, which requires proteolytic maturation to generate the bioactive species, IL-1α has minor bioactivity as the pro-peptide that is enhanced several fold by cleavage [143]. Caspase-5 was previously reported to cleave IL-1α [144], but our recent work revealed that IL-1α is not a substrate of the inflammatory caspases [54]. Our findings are congruent with other studies that identified granzyme B and calpains as the proteases that cleave IL-1α [145,146]. Recent work suggests that IL-1α can be activated downstream of GSDMD pore formation due to the influx of calcium, which activates calpains. Therefore, there is a possibility that IL-1α may be activated and released into the TME during pyroptosis [146]. However, the role of IL-1α in promoting antitumor immunity remains unclear and may be tumor specific, as studies suggest that it can act in both a pro- and antitumor manner [143,147]. It is worth noting that the depletion of IL-1β during tumor cell pyroptosis significantly attenuates antitumor immunity, suggesting that IL-1α may not play a major role in inducing antitumor immunity [66]. Further studies are needed to determine the impact of IL-1α and other chemokines and alarmins released during tumor cell pyroptosis in promoting an immunostimulatory environment that elicits antitumor immunity.

### 10.3. IL-18 Promotes Anti-tumor Immunity

In stark contrast to the development of immunotherapies that block IL-1ꞵ in cancer treatment, many cancer immunotherapies are being designed that promote IL-18 signaling due to its beneficial antitumor effects [118,126,127,148]. IL-18 was first recognized for its ability to induce IFNγ production in lymphocytes [128]. Further investigation demonstrated that IL-18 can activate NK and T cells [129,130]. Administration of recombinant IL-18 synergized with the costimulatory ligand B7.1 to induce melanoma regression [119] and early immunotherapies based on IL-18 have shown efficacy and the ability to generate antitumor responses. CAR T cells engineered to secrete IL-18, demonstrated improved anti-tumor efficacy [149]. Furthermore, a decoy-resistant IL-18 that is resistant to inhibition via the IL-18 binding protein showed anti-tumor effects as a monotherapy and in combination with ICB therapy [126]. However, a phase 2 randomized clinical trial, administering recombinant IL-18 for metastatic melanoma, failed to detect any clinical benefit [127].

While most studies indicate that IL-18 has proinflammatory antitumor effects, some studies suggest IL-18 could potentially be pro-tumorigenic. Much like IL-1ꞵ, IL-18 induces VEGF expression, which can promotes angiogenesis and stimulate tumorigenesis [132]. IL-18 may also play a role in metastasis, as it has been shown to enhance the migratory phenotype of melanoma cells [131]. This may be due to the fact that IL-18 induces endothelial adhesion molecules like VCAM and ICAM, which support tumor cell metastasis [133]. Future studies are needed to elucidate the precise role of IL-18 in tumor initiation, progression, and antitumor defense.

## 11. Concluding Remarks and Prospect

Pyroptosis is an immunogenic type of cell death that is emerging as a promising new strategy for immune-oncology applications. Pyroptosis induces antitumor immunity and sensitizes immunologically ‘cold’ tumors to immune checkpoint blockade therapy. Additionally, pyroptosis offers an alternative for chemotherapy resistant tumors that upregulate anti-apoptotic molecules. However, as promising as these finding are, several challenges still need to be overcome before pyroptosis-based therapies can be translated into meaningful clinical practices. First, the specific targeting of tumors using nanoparticles remains a challenge as all nanoparticles go to the liver [150]. Since cancer cells dampen the expression of Gasdermin proteins but normal cells, and in particular, cells in the gastrointestinal tract and hematopoietic cells maintain expression, targeting tumors and avoiding toxicity will be a challenge that needs to be overcome. Thus, mechanisms to ensure tissue-specific targeting need to be developed to limit the off-target toxicity of healthy tissues. Second, the immune microenvironment profoundly impacts therapeutic responses yet the precise DAMPS, cytokines, and chemokines that promote an immune-activating vs. immune-suppressive TME remains unclear. For example, are IL-1β and IL-18 released when pyroptosis is triggered via caspase-1-independent mechanism biologically active and what is the impact on therapeutic response? There are conflicting reports about the impact of inflammation on tumorigenesis, so key variables to consider when studying pyroptosis in cancer are the duration of inflammation, concentrations of released cytokines, and the cell types undergoing pyroptosis. The high levels of pyroptosis and cytokines released during acute treatment by malignant cells may induce strong antitumor responses, while low levels of pyroptosis by tumor-resident immune cells may promote tumor growth within the same cancer models. A systematic characterization of the immune landscape during pyroptosis at the protein and transcriptomic level will be critical for understanding and harnessing the antitumor properties of pyroptosis. Lastly, many of the pyroptosis-based therapies have been evaluated in only a handful of tumor models and further testing in a variety of tumor models is needed to determine the full scope of this treatment strategy. Excitingly, these pyroptosis-based therapies hold tremendous potential for igniting the immune systems to fight cancer.

## Figures and Tables

**Figure 1 cells-13-00346-f001:**
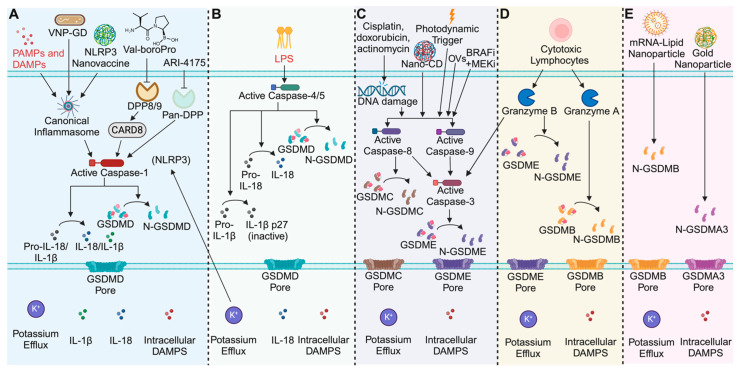
Overview of pyroptosis mechanisms and pyroptosis-based therapeutics. (**A**) Schematic of the canonical inflammasome pathway. Inflammasome oligomerization activates caspase-1, which cleaves and activates IL-1β, IL-18, and GSDMD. Therapeutic strategies like the synthesized glutathione (GSH)-responsive GSDMD protein cages tethered to an attenuated *Salmonella typhimurium* (VNP) strain (VNP-GD), the NLRP3 activating nanovaccine, and the DPP inhibitors Val-boroPro and ARI-4175 activate caspase-1 to treat cancer. (**B**) Schematic of the human non-canonical inflammasome pathway. Caspases-4/5 detect intracellular LPS and convert IL-18 and GSDMD to their active forms, while deactivating IL-1β. (**C**) Schematic of the apoptotic pathways that induce gasdermin-mediated cell death. DNA damage by chemotherapeutics, targeted small molecules, photodynamic triggers, oncolytic viruses (OVs) and the nanoparticle approach (NanoCD) induce apoptotic caspase activation. Caspases-3, and -8, cleave GSDME or GSDMC respectively to induce apoptosis. Caspases-8/9 can also activate caspase-3 to execute GSDME-mediated pyroptosis. (**D**) Cytotoxic lymphocytes inject granzymes into target cells. Granzyme B cleave GSDME and Granzyme A cleaves GSDMB to induce pyroptosis. (**E**) Next-generation nanoparticle approaches deliver activated gasdermin proteins to directly induce pyroptosis. Image created with Biorender.

**Table 1 cells-13-00346-t001:** Summary of pyroptosis inducing mechanisms, cancer models, and impact on adaptive immunity. Arrows indicate the direction in which immune cell levels changed following therapeutic intervention (↑ = increase, ↓ = decrease).

Agent	Mechanism	Tumor Models	Immune Phenotypes	Citations
Oncolytic Viruses (rAAV, HSV-1, VSV, CVB3, ORFV, rMV-Hu191)	CASP3/GSDME	Various cancer cell lines and models, (e.g., 4T1, EMT6 B16F10, ESCC, CT26)	T cells ↑, NK cells ↑, MDSCs ↓, Tregs ↓, and DC Maturation ↑	[97,98,99,100,101,102]
DNA Damaging Agents (cisplatin, doxorubicin)	CASP3/GSDME	Various cancer cell lines and models, (e.g., 4T1, EMT6 B16F10, GSDME^+^ HeLa, SH-SY5Y and MeWo cells)	T cells ↑, NK cells ↑, and Macrophage Infiltration ↑	[31,60,68,69,70,74]
BRAFi + MEKi	CASP3/GSDME	BRAF V600E Mut Melanoma	T cells ↑, TAMs ↓, MDSCs ↓	[67]
Cytotoxic Lymphocytes	Granzyme A/GSDMB, Granzyme B/GSDME	GSDME^+^ or GSDMB^+^ Cells (EMT6, SH-SY5Y, HeLa, 4T1, SW837, OE19, or SKCO1 cells. CT26,)	Innate Immune recruitment and Activation ↑, Bystander T cells ↑	[60,62,63,64]
Val-boroPro	CARD8, CASP1/GSDMD	Lymphoma, Myeloma, Acute Myeloid Leukemia	DC maturation ↑, Macrophage Activation ↑, T cell recruitment and activation ↑	[78,79,80,81,82,84]
Pan-DPP inhibitor, ARI-4175	CASP1	Hepatocellular Carcinoma	T cell Infiltration ↑	[90]
Decitabine + Cisplatin Liposome	CASP3/GSDME	4T1 Breast Cancer	DC Maturation ↑, CD8^+^ T cell Infiltration ↑	[106]
NanoCD	CASP3/GSDME	B16F10 Melanoma	T cell Infiltration and co-stimulation ↑, Cytokine Secretion ↑, Tregs ↓, Immunosuppressive CD206+ Macrophages↓	[107]
NP-GSDMA3	GSDMA3	4T1 Breast Cancer	T and NK cell infiltration ↑, M1 Macrophages ↑, Tregs ↓, M2 macrophages↓	[66]
VNP-GD + BAPTA-AM	CASP1/GSDMD + Ca^2+^ inhibition	4T1 Breast Cancer, B16F10 Melanoma, Inoperable Ovarian Cancer	DC Maturation ↑, CD8^+^ T cell infiltration ↑, Cytokine Secretion ↑	[113]
NLRP3 Nanovaccine	NLRP3/CASP1/GSDMD + TLR7/8 agonist	E.G7, B16F10 Melanoma, CT26 Colon Cancer	T cell infiltration ↑, Tregs ↓	[114]
A-C/NP	Photodynamic Trigger, CASP3, GSDME	4T1 Breast Cancer	T cell activation and infiltration↑	[120]
GSDMB-N LNP	GSDMB	4T1 breast cancer, B16F10 melanoma	T and NK cell infiltration and activation ↑, DCs maturation and activation ↑, Cytokine secretion ↑	[65]
IL-1β	Released from Canonical Inflammasomes via CASP1/GSDMD	Various models	T cell activation ↑, Dendritic cell maturation ↑ Plasma cell proliferation ↑, MDSC activation ↑, Negative immune feedback ↑, Angiogenesis ↑	[116,121,122,123,124,125]
IL-18	Released from Canonical and Non-canonical Inflammasomes via CASP1/4/5/GSDMD	Various models	IFNγ producing cells ↑, T and NK cell activation ↑, Angiogenesis↑	[126,127,128,129,130,131,132,133]

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
