# Peer review of "Harnessing Pyroptosis for Cancer Immunotherapy"

_cells, 2024, doi:10.3390/cells13040346_

Round 1

Reviewer 1 Report

Comments and Suggestions for Authors

The authors have written a very good review on the topic for cancer immunotherapy. They have summarized the main findings from basic research to clinical trials, mechanisms of pyroptosis and its biological consequences and functions in biological processes and antitumor immunity.

Generally speaking, the manuscript is well written, and suitable for publication.

In order to expand the topic and make the review more complete, and comprehensive, I do have one suggestion. Oncolytic virus (OV)-mediated therapy has been considered to be a novel type of cancer immunotherapy, and some of them induce pytoptosis, a type of immunogenic cell death (ICD), and thus elicit antitumor immunity. Under the Section of “Next-generation Approaches to Inducing Pyroptosis”, or as an independent Section before that, authors could/should add a new (sub)section of “oncolytic viruses induce pyroptosis for improved cancer immunotherapy” (or something like that). There are quite a few papers and important progress on the topic published in the last few years. For example,

1.      PMID: 37202386. Recombinant measles virus vaccine rMV-Hu191 exerts an oncolytic effect on esophageal squamous cell carcinoma via caspase-3/GSDME-mediated pyroptosis. Cell Death Discov. 2023;9:171.

2.      PMID: 36641456. Oncolytic Parapoxvirus induces Gasdermin E-mediated pyroptosis and activates antitumor immunity. Nat Commun. 2023;14:224.

3.      PMID: 36551691. Coxsackievirus Group B3 Has Oncolytic Activity against Colon Cancer through Gasdermin E-Mediated Pyroptosis. Cancers (Basel). 2022;14:6206.

4.      PMID: 36545949. A Dual-Responsive STAT3 Inhibitor Nanoprodrug Combined with Oncolytic Virus Elicits Synergistic Antitumor Immune Responses by Igniting Pyroptosis. Adv Mater. 2023;35:e2209379.

5.      PMID: 36096276. Vesicular stomatitis virus sensitizes immunologically cold tumors to checkpoint blockade by inducing pyroptosis. Biochim Biophys Acta Mol Basis Dis. 2022;1868:166538.

6.      PMID: 34887423. Strategies to package recombinant Adeno-Associated Virus expressing the N-terminal gasdermin domain for tumor treatment. Nat Commun. 2021;12:7155.

One minor issue:

1.      There are minor errors/misses in the following references:

Ref #3. Missing volume and page numbers; or article number (AN).

Ref #18. Same issue.

Ref #50. Is a peer-reviewed publication available now?

Ref #52. Updating: Nature. 2023;624:451-459.

Ref #53. Updating: 2023;624:442-450.

Ref #55. Any update?

Ref #88. Missing the article number.

Ref #89. The article number (replacing page numbers) should be, e10507, not e10507-n/a.

Ref #96. Missing the article number.

Ref #121. Missing the article number.

Ref #126. Missing the article number.

Reviewer 2 Report

Comments and Suggestions for Authors

This review by Bourne and Taabazuing presents an up-to-date understanding of pyroptosis and its role in enhancing immune responses and antitumor activities. It also describes the available therapies targeting pyroptosis in tumor cells to augment immune reactions. The review also explores the potential dual roles of IL1b and IL18, key cytokines released during pyroptosis.

To this extent, the possible release and role of IL1a should also be mentioned.

This review is a good contribution to the field, offering new insights and perspectives.

I have a few minor comments:

  1. 1. The discussion in Figure 1 should be sequential, progressing logically from panel A to D. Currently, it starts with panel B and then jumps to A, which disrupts the flow. Rearranging the figure or the discussion is needed.

  2.  
  3. 2. Paragraph numbering would greatly aid in the readability of the text.

  4.  
  5. 3. Highlighting the differences between immune cell-mediated pyroptosis and tumor cell-mediated pyroptosis in terms of antitumor activity would be beneficial.

  6.  
  7. 4. The section on Next-Generation approaches could be improved by adding subsections.

Round 2

Reviewer 1 Report

Comments and Suggestions for Authors

It is a great review.